# Analysing the Influence of Green Marketing Communication in Consumers’ Green Purchase Behaviour

**DOI:** 10.3390/ijerph20021356

**Published:** 2023-01-11

**Authors:** Elisabete Correia, Sara Sousa, Clara Viseu, Manuela Larguinho

**Affiliations:** 1Coimbra Business School|ISCAC and CEFAGE, 3040-316 Coimbra, Portugal; 2Coimbra Business School|ISCAC and CERNAS, 3040-316 Coimbra, Portugal; 3Coimbra Business School|ISCAC and CICF, 3040-316 Coimbra, Portugal; 4Coimbra Business School|ISCAC and CIMA, 3040-316 Coimbra, Portugal

**Keywords:** green marketing communication, green purchase behaviour, green products, Portugal

## Abstract

This study aims to explore whether consumers’ attention to companies’ green marketing communication influences their green purchase behaviour. It also analyses the importance of consumers’ characteristics, namely gender, education, and green attitudes, in their attention to companies’ green marketing communication. An online survey was carried out on the population residing in Portugal over 18 years of age, allowing us to collect 690 valid responses. Data analysis techniques including descriptive analyses, parametric and non-parametric tests, linear correlation, and regression analysis were used. The achieved results allow us to conclude that consumers are attentive to companies’ green marketing communication. A strong correlation between consumers’ attention to companies’ green marketing communication and green purchasing behaviour was identified. The results also confirm that individuals with higher educational levels and green attitudes and females are the most attentive to companies’ green marketing communication.

## 1. Introduction

Over recent decades, particularly since the industrial revolution, humanity’s rapidly growing consumption of natural resources has been causing many environmental problems worldwide, severely affecting biodiversity and threatening human well-being [1,2,3,4,5]. As individual consumers, our behaviour is not environmentally benign, being partly responsible for the severity of all these environmental problems [6,7]. Nevertheless, individuals’ growing perception that the planet is already reaching very high levels of pollution has contributed to the rise of an environmental protection “movement” [8,9]. Furthermore, both consumers and companies, particularly in more developed economies, are becoming increasingly aware of the urgent need to adopt more environmentally friendly consumer behaviour and production strategies [10,11]. 

Purchasing green products for daily consumption is a good example of an environmentally responsible behaviour, capable of minimising and solving many of the current environmental threats and, in recent years, it has been increasingly attracting the attention of both companies and consumers [12,13,14,15,16]. In addition to changing consumption patterns, by becoming greener, consumers also demand a more sustainable and environmentally responsible position from companies. These have a key role as they develop and promote the products we consume and thus contribute to shaping demand and its associated environmental impacts. Over the last few decades, the role of sustainability in business has gradually increased, and several companies have significantly contributed to the promotion of sustainable consumption [17,18,19]. Moreover, companies have increasingly adapted their activities within a more sustainable approach due to both stronger sustainability requirements and the implementation of new technologies aimed at improving the environmental, social, and economic business impacts [18]. Recently, Forbes [20] selected and ranked large companies that excel in sustainability combined with high revenues, offering several key insights into the success of large companies in sustainability. For companies that want to assume a “greener” position, to show consumers how committed they are to contributing to a more sustainable planet, and also to gain a differentiating factor concerning competitors, environmentally responsible marketing, also known as green marketing, represents an unquestionably essential tool [21]. According to Paço et al. [22], this tool is crucial, mainly when consumers show a low interest in certain products, including green ones.

If, on the one hand, research on green consumerism is evolving, justifying new research work that deepens knowledge in this area, green marketing is also an area for research due to its novelty and importance for environmental sustainability. The literature highlights the need to understand and promote consumer behaviour in green marketing and to identify the factors that influence green purchasing [23]. Tan et al. [24] highlight that green marketing is essential to change the environmental behaviour of consumers. According to the authors, companies should pay special attention to green communication, as it can increase customer confidence in a brand’s environmental commitment and thus positively influence green purchasing. Therefore, this research study was proposed to deepen the knowledge of how attentive consumers are, from the consumers’ perspective, to companies’ green marketing communication and how this influences consumers’ green purchase decisions. Since the 1990s, the economic growth of Portugal, a country that joined the European Economic Community (EEC) in 1986, has been considerably increasing, which has led to several environmental problems worsening. As a European Union (EU) member, Portugal has been receiving monetary funds to accompany the other EU members in adopting environmental protection measures. A clear increase in Portuguese concerns for the main environmental issues is observed, but little is known about consumers’ perceptions regarding green companies and marketing communication [25,26]. 

In this context, we propose to overcome this gap by studying Portuguese consumers’ attention to companies’ green marketing communication. To this end, an online survey was carried out on the population residing in Portugal over 18 years of age, allowing us to collect of a total of 704 responses, of which 690 were considered valid and incorporated in the analysis. 

The remainder of this paper is organised as follows. After an introduction to the theme, the second section reviews the scientific literature, explores some of the main concepts, and presents the research hypotheses. The following section is devoted to the methodology, with a detailed description of the questionnaire design, data collection, and sample profile. Then, the results are presented and discussed. Finally, in the last section, the main conclusions are presented along with policy and research implications.

## 2. Literature Review, Key Concepts, and Research Hypotheses

### 2.1. Green Marketing Communication 

Currently, there is no universal definition of green marketing. A detailed review of the scientific literature allows us to observe the existence of numerous studies explaining the significance of this concept [27]. There is also some freedom in using this term. It can be replaced with terms such as “environmental”, “ecological”, or “eco-” marketing [28]. According to Zafar et al. [29], green marketing is a marketing practice that raises environmental issues. Dangelico and Vocalelli [30] consider green marketing as a set of actions that aim to ensure that the product exchange has the smallest potential negative effect on the environment. Although the concept has evolved over the years, becoming more structured, we continue to find that different definitions of green marketing are more or less comprehensive [31]. Regardless of the term used and the approach adopted, green marketing is a crucial concept that aims to bring companies’ activities into a closer and more harmonious relationship with the environment. In this context, green marketing requires the inclusion of a broad range of activities and trends in marketing activities (e.g., modification of products, production processes, packaging, and labelling, as well as advertising strategies) aimed at satisfying human needs but with minimal environmental impacts (e.g., [32,33,34]). Polonsky [35] underlines that green marketing is the process of attempting to develop various strategies to target consumers who are more concerned for the environment. According to Cherian and Jacob [36], companies use green marketing to increase the awareness of customers and show them that the company seeks to contribute to solving environmental problems. Green marketing has become one of the key developments in modern business, which is more applied in developed countries than in lower- and middle-income countries [37].

One of the most decisive factors for the success of a socially responsible company (a company with the social obligation to carry out the policy in making decisions and acting in accordance with the accepted values of society) is its efficient communication with consumers, and green marketing, as a part of socially responsible communication, can positively influence consumers’ purchase behaviours [28,38]. Companies have been communicating the environmentally friendly characteristics of their products, leading to the growth of a significant segment of well-informed green consumers who criticise producers about the type of communication delivered [39]. Communication is one of the fundamental instruments to support organisational change [40]. 

In this study, it is considered that green marketing communication incorporates a broad range of activities that aims to attract consumers’ attention both to the characteristics of the green products and to the company’s broad range of environmentally friendly activities [33]. In this context, it is expected that efficient green marketing communication will have a positive impact on consumers’ behaviour by encouraging them to buy environmentally friendly products produced by companies that develop their activities with respect for the environment [41,42].

Companies’ green marketing communication may be carried out in different ways, namely through advertising, corporate public relations, visual identifications, green labels and packaging, and sustainability reports, among others [24,43]. Furthermore, it can be carried out through a broad range of channels, namely social networks, websites, newspapers, brochures, commercials on television, and magazines [44,45]. All these communication channels convey messages to persuade customers that their decision to be greener benefits both the environment and their own health [46,47].

Messages used for environmental communication are both verbal and non-verbal. For instance, green labels, also known as environmental or eco-labels, are used to convey, through images or texts, messages of environmental friendliness, giving consumers information about green manufacturing, packaging, and waste management activities [48]. These labels are often used by companies to differentiate their products, and act as a guide for consumers to choose environmentally friendly products [49,50]. According to Rashid [51], awareness of eco-labels has a positive effect on the knowledge of a green product and consumers’ intention to purchase. Other studies indicate that although the functions of labels are recognised by some consumers, this does not automatically lead them to green purchasing decisions [52]. As D’Souza [53] stressed, little is understood about the effect of label information on a consumer’s intention and decision to purchase environmentally friendly products. Companies have used other instruments to communicate the environmentally friendly characteristics of their products and other activities and initiatives that demonstrate their commitment to the environment. For example, advertising deserves some attention in the literature, with several studies analysing, among other aspects, its impact on green purchasing intention or green purchasing behaviour [22,30]. For example, Paço et al. [22] analysed the impact of advertising on purchasing behaviour; however, the results of their study were not conclusive.

### 2.2. Green Consumers

Having deepened the concept of green marketing communication, it is now important to delve into another closely associated concept: the green consumer. Green marketing is aimed at all consumers, with the aim that they become increasingly green and value the attitude of companies as environmentally friendly, choosing to buy their products. Although there are different definitions of green consumer [54], one of the most adopted and agreed upon is given by Hailes [55], who defines this kind of consumer as one who associates the act of buying or consuming products with the possibility of acting in accordance with environmental protection. To Dagher et al. [56], a green consumer is an environmentally friendly consumer who is trying to contribute to solving environmental problems through her/his green purchasing behaviour. For green consumers, the job of protecting the environment should not be left to others (e.g., government, business, environmentalists, and scientists); they, as consumers, can also play an important part and make a difference [57]. The green consumer knows he/she is actively protecting the environment by refusing to buy products that harm the environment during production, use, or final disposal, consume much energy, and have non-recyclable packaging. Biswas and Roy [58] stressed that green products entail several potential benefits to the environment as they are made of environmentally friendly resources, have resource-conservation potential, can be recycled, and have the least environmental impact at all stages of their lifecycle. Following this definition, in this study, the products of interest are: green cosmetic and hygiene products; green detergents and other cleaning products; clothes produced with natural and/or recycled materials; food from organic farming without using any pesticides; products made from recycled materials such as glass and paper; electrical appliances with high energy efficiency; sustainable building materials and furniture; recycled mobile phones; electric/hybrid vehicles whose use emits less carbon dioxide; and eco-hotels (hotels that adopt green practices and/or whose buildings respect the environment). Nowadays, several companies in Portugal produce, sell, and/or distribute these green products, developing marketing strategies to show consumers their concern for the environment. However, the following question arises: *Are consumers paying attention to companies’ green marketing communication? If so, does it influence consumers’ green purchasing decisions?* Hence, it was intended to explore to what extent consumers pay attention to companies’ communication, that includes green messages, either to promote its products or the company itself, and whether these green messages influence consumers’ green purchasing.

Considering some conflicting results regarding the influence of green communication on green purchasing behaviour, it was proposed to verify the following research hypothesis:

**Hypothesis 1 (H1).** *Consumers’ attention to companies’ green marketing communication influences their green purchasing behaviour*.

### 2.3. Consumers’ Characteristics

In this section, we review some scientific literature that explores the importance of consumers’ characteristics, in particular gender, attitudes, and education, in their attention to companies’ green marketing communication.

#### 2.3.1. Gender

In recent decades, several researchers have stressed the importance and the need to consider gender when studying individuals’ environmental behaviour, since it may influence important issues, namely attitudes, beliefs, concerns, opinions, and behaviours [59,60,61]. In many research studies, the achieved results underline that, in comparison to men, women are more concerned with the environment and adopt more pro-environmental behaviours (e.g., [60,62,63,64,65,66]). Zelezny et al. [67] conducted an international survey covering 14 countries regarding gender differences in pro-environmental attitudes and behaviours. They found a significant difference between genders, concluding that women present stronger pro-environmental attitudes and behaviours than men, and higher levels of socialisation and social responsibility. In other studies, the authors suggest that women declare to carry out more pro-environmental actions when asked about daily behaviours such as resource conservation, recycling, and transport use [68,69]. Balzekiene and Telesiene [70] distinguish the private from the public sphere and conclude that active private sphere environmental behaviour is more common among women.

Regarding green purchase behaviour specifically, several authors stress the importance and the influence of the consumers’ gender in their green purchasing decisions (e.g., [71,72]). Ureña et al. [73] compared Spanish men and women and observed that women are more prone to purchase green products. Lee [8] explored gender differences in Hong Kong adolescent consumers’ green purchasing behaviour and concluded that in 6010 adolescents, females scored significantly higher in green purchasing behaviour than males. In a Croatian case study, Radman [74] also concluded that women purchase green products more often than men. Han et al. [75] found that women showed higher intention to use a green hotel and pay premium prices for it. Chekima et al. [76] concluded that Malaysian highly educated women showed higher green purchase intentions.

Nevertheless, despite all these findings observing that women tend to act more ecologically than men in terms of sustainable consumption, we cannot generalise and conclude that the same is observed worldwide. Gilg et al. [77] found no gender effect in the UK; Chen and Chai [78] did not observe any effect of this type in Malaysia, and the same holds for Zhu et al. [79] for China. In the case of Diamantopoulos et al. [68], these authors reported several studies concluding that men have higher environmental knowledge about environmental issues than women and act accordingly. Additionally, Mostafa [80] found that Egyptian men are more concerned about the environment and more prone to purchase green products than women. Considering that many studies conclude that there are gender differences regarding behaviours, or more favourable intentions for purchasing green products, it is expected that there are gender differences regarding consumers’ attention to companies’ green marketing communication. In this context, it was proposed to verify whether:

**Hypothesis 2 (H2).** *Consumers’ gender influences their attention to companies’ green marketing communication*.

#### 2.3.2. Green Attitudes

The literature provides several definitions for environmental or green attitudes. Milfont and Duckitt [81] refer to this concept as a psychological tendency expressed by evaluating the natural environment with some degree of favour or disfavour. In Gifford and Sussman [82] (p. 65–66), green attitudes are defined as “concern for the environment or caring about environmental issues (sometimes referred to as pro-environmental attitudes)”. According to Ugulu et al. [83], it is essential to study and act on possible negative environmental attitudes since it is observed that individuals with negative attitudes towards the environment underestimate environmental problems and tend not to adopt environmentally friendly behaviours. 

Particularly regarding green purchase behaviour, several authors proposed to explore the relationship between individuals’ attitudes, intentions, and actual green purchase behaviour (e.g., [84,85,86,87,88]). For instance, Urban and Kaiser [89] found that the environmental attitudes of people with low, moderate, and high propensities for green consumption differed systematically in the 28 European countries analysed in their research study. With these findings, the authors could associate people’s environmental attitudes (i.e., their commitment to protecting the environment) with their protective engagement and hence corroborate a generalisable positive relationship between environmental attitude and engagement in environmentally protective behaviour across a relatively large pool of countries. Nevertheless, some studies observed a weak correlation between the expressed positive attitude of consumers towards green purchases and their actual purchase behaviour, known as the attitude–behaviour gap (e.g., [90,91,92]). Some theories justify this gap with the existence of contextual factors that strongly affect the attitude–behaviour relation (e.g., [93,94,95]). 

In this study, we intended to analyse the relationship of the individual’s environmental attitudes, not directly with the green purchase behaviour, as this has already been carried out, but with the individual’s attention to companies’ green marketing communication. In this context, the following research hypothesis was proposed:

**Hypothesis 3 (H3).** *Consumers’ green attitudes influence their attention to companies’ green marketing communication*.

#### 2.3.3. Education

When dealing with different environmental problems, it is important to raise people’s awareness to act and mitigate the current environmental problems. Environmental education is one of the most effective tools for teaching and training people to have environmental responsibility and adopt more pro-environmental behaviours [96]. According to Salequzzaman and Stocker [97], environmental education helps people become aware of the consequences of their actions, provides information to help solve environmental problems, and develops the human capacity to solve and prevent environmental aggressions. Environmental education involves the interchange of knowledge to build values, attitudes, and skills that prepare individuals to collaboratively undertake positive environmental action [98]. However, to be effective, education cannot be a unidirectional transfer of information; it must instead be a collective construction that includes the social reality and the traditional knowledge of local communities [99]. Lozano [100] found that an individual’s environmental concerns and behaviour increase with higher education. According to Schlegelmilch et al. [101], highly educated individuals seem to possess a higher level of environmental knowledge and are motivated to engage in environmentally responsible behaviours. Individuals with high education levels were also identified by Straughan and Roberts [102] to be more likely to act green. 

In this context, it is expected that these highly educated individuals are more attentive to companies’ green marketing communication, and so the following research hypothesis was proposed:

**Hypothesis 4 (H4).** *Consumers’ education level influences their attention to companies’ green marketing communication*. 

Based on the literature review above, Figure 1 sets out the research model that illustrates the hypotheses discussed previously.

## 3. Research Methodology 

### 3.1. Questionnaire Design and Survey Procedures

The data collection method was a survey. It took the form of a self-administered questionnaire. After an extensive literature review on consumers’ green purchasing behaviour, a significant set of questions was selected to gather essential information on respondents’ behaviour. This provisional questionnaire version was subject to the “think aloud” technique, a valuable qualitative research tool, mainly consisting of cognitive interviewing. Participants are asked to verbalise all thoughts that would normally be silent, allowing direct data to be collected on the respondents’ ongoing thinking process as they answer the questions [103,104,105]. This procedure took place in Coimbra on 28 February 2021. Ten individuals participated, whose testimonies allowed us to build the final and definitive version of the questionnaire, with the reformulation of some questions, making them simpler and clearer, and with the elimination of some questions, since most participants complained about the high number of questions.

The questionnaire was composed of two main sections: the first included several questions to gather information on respondents’ demographic characteristics, namely age, gender, education, and occupation; and the second included different questions on key issues in respondents’ green purchasing behaviour, namely attitude, green purchase behaviour, and attention to companies´ environmental marketing. Based on an extensive literature review (e.g., [22,106,107,108]), a selection of several statements (see Appendix A) was presented to respondents who were asked to rate their agreement using a five-point Likert scale (1—strongly disagree, 2—disagree, 3—neither agree nor disagree, 4—agree, 5—strongly agree).

### 3.2. Data Collection and Statistical Analysis

The questionnaire was distributed to participants online, on various social media platforms, such as Facebook and Twitter, WhatsApp, and email, over a month, from 1 to 31 March 2021, allowing us to collect of a total of 704 responses, of which 690 were considered valid and incorporated in the analysis. The decision to collect data online was due to several reasons, of which we highlight that the majority of the residents in Portugal were confined in their residences during March as a measure to combat the spread of the virus COVID-19. Therefore, rather than a choice, it was an imposition. In addition, the use of the internet presents several advantages over other methods, namely face-to-face and telephone interviews, allowing the collection of a large number of responses in a short period of time and at insignificant costs. It should also be noted that comparative experimental studies have shown that the differences in the results of studies conducted through direct or telephone interviews or the internet are insignificant [109,110,111]. After collection, the data were statistically analysed using the statistical software IBM SPSS version 27. 

Various statistical methodologies were applied to analyse this study´s research hypotheses. Descriptive statistics were used to explore the variable green marketing communication. A simple regression model was obtained to analyse the impact of the green marketing communication on green purchase behaviour. To study the influence of gender and education on attention to companies’ green marketing communication, in an individual way, parametric and non-parametric tests were applied. Finally, a multiple regression model was estimated to explore the importance of consumers’ characteristics, in particular gender, attitudes, and education, on their attention to companies’ green marketing communication.

### 3.3. Sample Profile

All respondents participated in this study voluntarily and were 18 or over, assuming that they have income and are buyers who are independent in making purchasing decisions. The respondents’ sociodemographic profile is presented in Table 1. As can be seen, 66.2% of the respondents are female; 61.8% are aged between 18 and 31 years old; 48% have higher education, of which 14.5 % have a master’s degree or Ph.D. 

## 4. Discussion

This research focuses on the attention to companies’ green marketing communication (latent variable with seven items) and the relationship of this variable with consumers’ green purchasing behaviour (latent variable with nine items), consumers’ green attitudes (latent variable with three items) as well as with gender and education level. Cronbach’s alpha was used to assess the internal reliability of the latent variables. The resulting values are 0.924 for attitude, 0.925 for behaviour, and 0.915 for attention; therefore, all the variables present very good reliability.

In one of the questionnaire’s sections, respondents were presented with different statements (see Appendix A) regarding their attention towards companies’ green marketing communication. They were asked to express their degree of agreement. The results are presented in Figure 2 and allow us to conclude that a large percentage of respondents show: paying attention to advertising messages on environmental protection; appreciating brands or companies that have environmental certifications and labels; responding favourably to brands that use messages about environmental protection in their advertising; preferring products from companies that adopt a responsible attitude towards the environment, to the detriment of others; and supporting ways of promoting products through environmentally friendly instruments. We highlight the high number of respondents that agree (30.9%) and strongly agree (51.5%) that it is important that companies provide/disclose more information about the environmental characteristics of their products and production methods. On the other hand, a considerable number of respondents (50.7%) do not pay close attention to product labels and tags with all the information about the environmental impacts, or are indifferent/neutral. This result stands out from the others since, in comparative terms with the remaining statements, it presents the highest percentage of respondents who strongly disagree, disagree, or assume a neutral position.

In order to examine the impact of the consumers’ attention to companies’ green marketing communication on their green purchasing behaviour, a simple regression model was performed. The results are presented in Table 2.

The variable consumers’ attention is significant to the model (t(640) = 0.821, *p*-value = 0.000), that is, variation in consumers’ attention implies a significant variation in the green purchasing behaviour. The coefficient of determination is R^2^ = 0.45, and the model is globally relevant (F(1, 640) = 523.974, *p*-value = 0.000). It can be concluded that 45% of the variation in green purchasing behaviour can be explained by the consumers’ attention. Therefore, hypothesis 1 can be supported according to the present evidence. 

Various statistical techniques were used to analyse the research hypotheses H2, H3, and H4. Firstly, the influence of consumers’ gender on their attention to companies’ green marketing communication was analysed using a parametric test. The test results shown in Table 3, more specifically, t(661) = 5.010 and *p*-value = 0.000, indicate the existence of significant differences considering gender. The results allow us to conclude that women are significantly more attentive to companies’ green marketing communication (female mean = 27.85, SD = 5.43) than men are (male mean = 25.59, SD = 5.61). Then, the results suggest the validation of H2. 

To compare the attention to companies’ green marketing communication between consumers with different educational levels, the non-parametric Kruskal–Wallis test was applied. This non-parametric test was selected because the dependent variable does not follow a normal distribution in each group of consumers. As Table 4 presents, there is a significant effect of educational level on attention to companies’ green marketing communication (*p*-value = 0.014 < 0.05), supporting H4: *Consumers’ education level influences their attention to companies’ green marketing communication*.

Table 5 shows the differences between two pairs of groups. Significant differences were found, at the 1% significance level (*p*-value < 0.01), in consumers’ attention to companies’ green marketing communication between the two groups, the consumers with secondary school or less and those with a master’s degree or Ph.D. At the 10% significance level, we found significant differences between consumers with secondary school or less and graduate consumers.

To explore the importance of consumers’ characteristics, in particular gender, attitudes, and education, in their attention to companies’ green marketing communication, a multiple regression model was estimated (Table 6). A significant regression was found (F(4,652) = 48.229; *p*-value = 0.000), with an R^2^ = 0.228. Gender (β = 0.198, *p*-value = 0.008), attitudes (β = 0.442, *p*-value = 0.00), and education level (graduate: β = 0.145, *p*-value = 0.059 and master’s or Ph.D.: β = 0208, *p*-value = 0.043) were found be positively and significantly associated with attention to companies’ green marketing communication. Hence, it may be concluded that women are more attentive to companies’ green marketing communication than men, supposing they have the same attitudes and independent of the education level. Consumers with a higher level of education tend to be more attentive to companies’ green marketing communication, also assuming they have the same attitudes and independent of gender. Therefore, hypotheses 2, 3, and 4 are supported.

Consumers’ attention to companies’ green marketing communication is directed, in particular, to information that companies provide about the characteristics of products and their production methods. Consumers consider this issue very important (*Statement 7*, in Appendix A), so companies should consider this aspect in their communication strategies. A key element in these strategies is, for example, defining the most appropriate communication channels. One of the channels is the internet, either through social networks or corporate websites, as these media have many advantages: they allow information to be conveyed in many ways (text, images, moving images, sound, interaction), and they can be strategic by making information more appealing and enabling interaction with consumers’ [112]. However, Sharma [23] highlights the risks associated with this channel. According to Sharma, these risks have restricted the purchase of green products in some countries. Thus, marketing professionals must consider the benefits and the risks associated with using these means. As stressed in other studies (e.g., [113]), it would be expected that labels and packaging would also be prominent and would be considered important for consumers as green marketing tools with the potential to influence consumer behaviour. However, in this study, respondents attach little importance or are indifferent/neutral to this issue (*Statement 4*, in Appendix A). This result could be related to the fact that consumers may have some distrust of labels. Labels, seals, and certificates are sometimes used only to attract consumers, often with misleading claims of environmental benefits associated with a product, also known as greenwashing [114]. Furthermore, no eco-label is recognised as credible and constitutes an international standard, which can also contribute to this distrust [113].

The results of this study reveal that consumers with more pro-environmental attitudes are also more receptive to companies’ communication regarding environmental issues. In a situation where consumers are more concerned about the environment, green marketing has proven effective as the message targets individuals already concerned about environmental issues [115]. Although communication may be critical to raise consumers’ environmental awareness when they do not have many environmental concerns [36], more significant efforts are required by companies to convince their audience. Such communication may be a valuable tool to target individuals who already have a positive attitude towards the environment and who may be more predisposed to this communication, reinforcing their positive attitude [116]. Rahman et al. [113] highlight the role of marketers in raising consumers’ awareness and promoting knowledge regarding green products. The literature points out that individuals with higher educational levels have higher environmental knowledge and are more encouraged to adopt responsible environmental behaviours [101]. The results of our study reveal that these individuals are also more attentive to companies’ green marketing communication. Furthermore, there are also gender differences regarding companies’ green marketing communication. As highlighted by some studies regarding green purchase behaviour (e.g., [8,73]), compared to men, women have more environmentally friendly behaviours and are more attentive to green marketing communication. Companies’ communication strategies must therefore take this population segment into account. Additionally, according to the results, receptivity and attention to companies’ green marketing communication are correlated with the consumers’ green purchase behaviour. Chen et al. [41] and Paço et al. [22] have also proven this relationship. Based on these results, companies should consider green marketing communication as a key tool to promote green purchases.

## 5. Conclusions and Future Research

Green purchasing is an environmentally responsible behaviour characterised by advocating for nature and protecting the environment. It is an effective way to solve environmental problems and has attracted companies’ and consumers’ attention in recent years [2,12,14,15,16]. As a result, for the last few years, consumers have been expressing a growing interest in purchasing green products. On the other hand, companies, academics, and researchers are revealing a growing interest in exploring the marketing impact in promoting a more balanced environment [117,118]. Marketing elements, namely product, price, distribution, and marketing communication, can be designed and applied in less harmful ways to the natural environment [119].

This study proposed to deepen the research on consumers’ attention to companies’ green marketing communication and how this determines consumers’ green purchase behaviour. As Bailey et al. [116] stressed, consumers do react differently to green marketing communication. Our results allowed us to validate all the presented research hypotheses: a positive correlation exists between consumers’ attention to companies’ green marketing communication and their green purchase behaviour. Moreover, results also proved that consumers presenting green attitudes, and who are more educated and female, are more attentive to companies’ green marketing communication. The findings reinforce the argument that companies benefit from communicating with their stakeholders about their commitment to sustainability and disseminating information integrating these concerns into their policies, business practices, and products [120,121]. In addition to gaining the support of these stakeholders, communication not only contributes to increasing transparency and enhancing reputation and legitimacy [122] but also seems to influence behaviour concerning purchases of green products, translating into economic benefits.

The present study contains some limitations. First, the sample is not entirely random and the survey took place online, thus excluding some members of the population who lack internet access. Secondly, the survey was distributed on social networks, so the sample may also represent individuals who use social networks more, which may explain specific characteristics of the sample (e.g., age structure, educational structure). Therefore, we recommend some caution as regards any generalisation of these results.

This research can be further expanded, deepened, and improved in the future, with the inclusion of more variables, extended to other countries with different degrees of development and from different continents, and address the behaviour of both companies and consumers towards specific green products. For example, it would be interesting to verify whether the results obtained in this study are similar for durable products (such as household appliances) and non-durable products (such as clothing), or services (such as eco-hotels). Another aspect to be analysed may be whether the market channels (online vs. offline) influence consumers’ choices concerning the purchase of green products.

## Figures and Tables

**Figure 1 ijerph-20-01356-f001:**
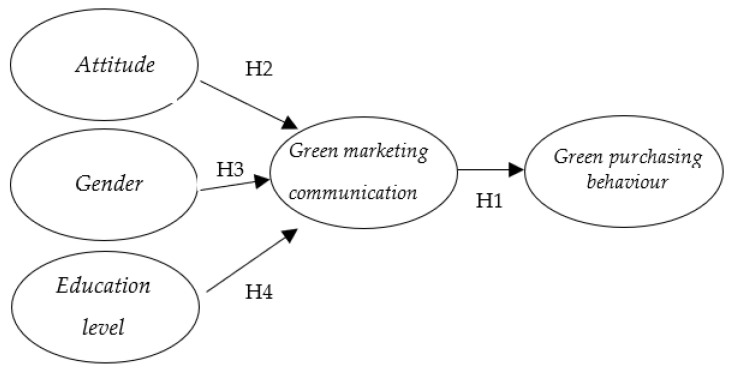
Proposed research model.

**Figure 2 ijerph-20-01356-f002:**
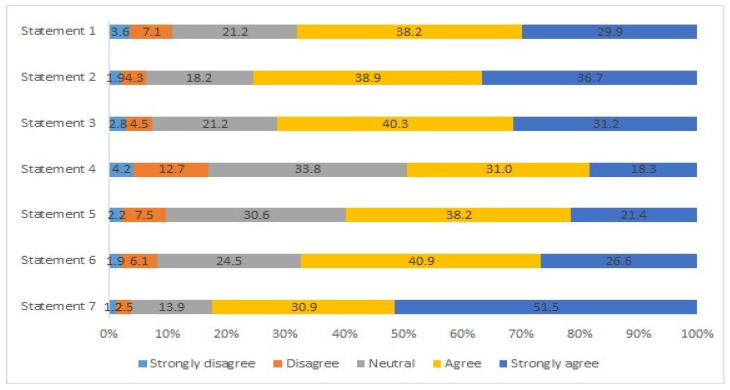
Consumers’ attention to companies’ green marketing communication. Source: Authors’ own elaboration. Note: See Table A1, in Appendix A. Statements: 1: I tend to pay attention to advertising messages that talk about environmental protection; 2: I appreciate brands/companies that have environmental certifications and labels; 3: I respond favourably to brands that use messages about environmental protection in their advertising; 4: I pay close attention to product labels and tags with all the information about their environmental impacts; 5: When purchasing, I prefer products from companies that adopt a responsible attitude towards the environment, to the detriment of others; 6: I support ways of promoting products through environmentally friendly instruments; 7: I consider it important that companies provide/disclose more information about the environmental characteristics of their products and production methods.

**Table 1 ijerph-20-01356-t001:** Respondents’ demographic profile.

Gender	Absolute Values	Percentage Values (%)
Male	233	33.8
Female	457	66.2
Missing	0	0
Total	690	100
**Age**	**Absolute Values**	**Percentage Values (%)**
18–21	289	41.9
22–31	137	19.9
32–51	177	25.7
52–77	75	10.9
Missing	12	1.7
Total	690	100
**School level**	**Absolute Values**	**Percentage Values (%)**
Up to secondary	359	52.0
Graduate	231	33.5
Master’s or Ph.D.	100	14.5
Missing	0	0
Total	690	100

Source: Authors’ own elaboration.

**Table 2 ijerph-20-01356-t002:** Simple regression results.

	Unstandardised Coefficients	Standardised Coefficients (Beta)	t-Statistics	*p*-Value
**Constant**	10.102		10.159	0.000 ***
**Attention**	0.821	0.671	22.890	0.000 ***

Dependent variable: Green purchasing behaviour. *** Significant at 1% level. Source: Authors’ own elaboration.

**Table 3 ijerph-20-01356-t003:** Results of the *t*-test for the mean difference in consumers’ attention to companies’ green marketing communication, exploring gender differences.

		Levene’s Test for Equality of Variances		*t*-Test for Equality of Means			
		F	*p*-Value	T	df	*p*-Value	Mean Difference
**Attention to companies** **’** **green marketing communication**	Equal variance assumed Equal variance not assumed	0.495	0.482	5.010 4.956	661 430.405	0.000 *** 0.000	2.263 2.263

*** Significant at 1% level. Source: Authors’ own elaboration.

**Table 4 ijerph-20-01356-t004:** Results of the Kruskal–Wallis test to compare consumers’ attention to companies’ green marketing communication, exploring education level.

School Level	n	Rank Mean	H Kruskal–Wallis	df	*p*-Value
**Up to secondary**	347	313.68	8.563	2	0.014 **
**Graduate**	219	342.34			
**Master’s or Ph.D.**	97	374.20			

** Significant at 5% level. Source: Authors’ own elaboration.

**Table 5 ijerph-20-01356-t005:** Results of the multiple comparisons to the different groups of consumers.

Comparison Group	Reference Group	Mann–Whitney U Test	*p*-Value
**Up to secondary**	**Graduate**	−1.738	0.082 *
	**Master’s or Ph.D.**	−2.758	0.006 ***
**Graduate**	**Master’s or Ph.D.**	−1.367	0.172

* Difference is significant at the 10% level; *** difference is significant at the 1% level. Source: Authors’ own elaboration.

**Table 6 ijerph-20-01356-t006:** Multiple regression results.

	Unstandardised Coefficients	Standardised Coefficients (Beta)	t-Statistics	*p*-Value
**Constant**	−0.215		−3.214	0.001 ***
**Gender**	0.198	0.093	2.640	0.008 ***
**Attitudes**	0.442	0.440	12.487	0.000 ***
**Graduate**	0.145	0.068	1.891	0.059 *
**Master’s or Ph.D.**	0.208	0.073	2.025	0.043 **

Dependent variable: Attention to companies’ green marketing communication. *** Significant at 1% level; ** significant at 5% level; * significant at 10% level. Source: Authors’ own elaboration.

## Data Availability

The scientific data will be made available by the authors whenever interest is expressed.

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
