# Peer review of "Analysing the Influence of Green Marketing Communication in Consumers’ Green Purchase Behaviour"

_ijerph, 2023, doi:10.3390/ijerph20021356_

Round 1

Reviewer 1 Report

The article "Companies` Green Marketing Communication: How Attentive are Consumers?" Does It Influence Green Purchase Behaviour?” I do not recommend printing, because in my opinion the authors made several serious mistakes in their scientific work. Especially:

1)      They used outdated professional literature during the search. In their search, they used only approximately 27 % of resources published in the last 5 years, and overall only 42% of resources are less than 10 years old. 58% of resources are older than 10 years. This has caused the authors to state ideas that have been overcome by current knowledge. It is hard to believe that, for example, this conclusion has not yet been overcome: The author also found that men in Hong Kong are more likely to search for information on green products and to purchase such products, particularly health food products. (Ling-Yee 1997)

2)      The authors did not focus on defining the content of marketing communication. Somewhat unsystematically, they chose only a few from a wide range of marketing communication activities, without justifying why these in particular. Moreover, the statements that were used in the questionnaire (especially statements 5 and 7) cannot be used to reveal consumers' attention to marketing communication activities. For example, statement 7 explores the importance of providing information but not paying attention to that information (they may consider it important to provide that information, but that doesn't mean the respondent actually pays attention to it).

3)      A big problem is the selection of a sample of respondents. Individual groups of respondents were unevenly represented (2/3 women, 52% of respondents under 31 years of age, more than half of the respondents were in the "until secondary" category). Probably the authors obtained the answers without random selection and also without respecting the population structure of Portugal. The conclusions cannot then be generalized at all. It is valid only in the specially created sample of respondents.

4)      For regression analysis “the coefficient of determination is R2 = 0.45.” Is it a large enough coefficient to accept the proposed model? After all, only "45% of variation in green purchasing behavior can be explained through the consumers' attention".

5)      Also, some conclusions made in the discussion are problematic, e.g. “….. that labels and packaging would have another prominence and would be considered important for consumers as green marketing tools with the potential to influence consumer`s behavior. However, in this study, respondents attach little importance to this issue ….” But according to Figure 1, half of respondents agreed with the statement "I pay close attention to product labels and tags with all the information about their environmental impacts" and almost 34% of respondents did not answer the question...

Author Response

To Reviewer 1:

First of all, we would like to thank you for all your contributions that allow us to considerably improve our research paper. Below, we present each of the changes made according to your comments:

Comment: They used outdated professional literature during the search. In their search, they used only approximately 27 % of resources published in the last 5 years, and overall only 42% of resources are less than 10 years old. 58% of resources are older than 10years. This has caused the authors to state ideas that have been overcome by current knowledge. It is hard to believe that, for example, this conclusion has not yet been overcome: The author also found that men in Hong Kong are more likely to search for information on green products and to purchase such products, particularly health food products. (Ling-Yee 1997)

Answer: Thank you very much for the comment and as such, we proceeded to eliminate some older bibliographic references, inserting much more recent ones. Below, we list some of the current references inserted throughout the text:

Chan, H.-W.; Pong, V. and. Tam K.-P (2019) Environ. Behav., 51 (1) (2019), pp. 81-108, 10.1177/0013916517735149

DÄ…browska A., JanoÅ›-KresÅ‚o M. (2018). Collaborative consumption as a manifestation of sustainable consumption. Problemy Zarzadzania-Manage. Issues. 2018;16(3):132–149. 

Dangelico, R. M., & Vocalelli, D. (2017). “Green Marketing”: An analysis of definitions, strategy steps, and tools through a systematic review of the literature. Journal of Cleaner production, 165, 1263-1279.

Ganganaboina, A.Y., & Sana, R. (2017). Communication of Green Marketing Strategies for Creating Consumer Awareness : A study of grocery retail sector in Sweden.

Hansmann R., Laurenti R., Mehdi T., Binder C.R. (2020). Determinants of pro-environmental behavior: A comparison of university students and staff from diverse faculties at a Swiss University. J. Clean. Prod. 2020;268:121864. doi: 10.1016/j.jclepro.2020.121864.

Li Y, Wang B, Saechang O. (2022). Is Female a More Pro-Environmental Gender? Evidence from China. Int J Environ Res Public Health. 2022 Jun 29;19(13):8002. doi: 10.3390/ijerph19138002. PMID: 35805661; PMCID: PMC9266259.

Majeed, M. U., Aslam, S., Murtaza, S. A., Attila, S., & Molnár, E. (2022). Green Marketing Approaches and Their Impact on Green Purchase Intentions: Mediating Role of Green Brand Image and Consumer Beliefs towards the Environment. Sustainability, 14(18), 11703.

Migheli, M. (2021). Green purchasing: the effect of parenthood and gender. Environ Dev Sustain 23, 10576–10600 (2021). https://doi.org/10.1007/s10668-020-01073-6

Ndofirepi, T.M.; Matema, S.C. (2019). Exploring Green Purchasing Behaviour among College Students in a Developing Economy. South. Afr. Bus. Rev. 2019, 23, 25.

Nunes, C.Q. (2019). Green Marketing and the Conscious Consumers   in Portugal. Master's dissertation. Lisbon: ISCTE-IUL. Available at www:http://hdl.handle.net/10071/19288

Paço, A., Shiel, C., & Alves, H. (2019). A new model for testing green consumer behaviour. Journal of cleaner production, 207, 998-1006. https://doi.org/10.1016/j.jclepro.2018.10.105

Shabbir, Muhammad & Bait Ali Sulaiman, Mohammed Ali & Sulaiman, Ali & Al-kumaim, Nabil & Mahmood, Arshad & Abbas, Mazhar. (2020). Green Marketing Approaches and Their Impact on Consumer Behavior towards the Environment-A Study from the UAE. Sustainability. 12. 10.3390/su12218977.

Silvi M., Padilla E. (2021). Pro-environmental behavior: Social norms, intrinsic motivation and external conditions. Environ. Policy Gov. 2021;31:619–632. doi: 10.1002/eet.1960.

Sreen, N., Purbey, S., Sadarangani, P., (2018). Impact of culture, behavior and gender on green purchase intention. J. Retailing Consum. Serv. 41, 177–189. https://doi.org/10.1016/j.jretconser.2017.12.002

Tan, Z.; Sadiq, B.; Bashir, T.; Mahmood, H.; Rasool, Y. (2022). Investigating the Impact of Green Marketing Components on Purchase Intention: The Mediating Role of Brand Image and Brand Trust. Sustainability 2022, 14, 5939. https://doi.org/10.3390/su14105939

Wiedmann, T., Lenzen, M., Keyßer, L.T. et al. (2020). Scientists’ warning on affluence. Nat Commun 11, 3107 (2020). https://doi.org/10.1038/s41467-020-16941-y

Zafar, Sohaib, Atif Aziz, & Muhammad Hainf. (2020). Young Consumer Green Purchase Behavior. International Journal of Marketing Research Innovation 4(1): 1–12. https://doi.org/10.46281/ijmri.v4i1.493

Comment: The authors did not focus on defining the content of marketing communication. Somewhat unsystematically, they chose only a few from a wide range of marketing communication activities, without justifying why these in particular. Moreover, the statements that were used in the questionnaire (especially statements 5 and 7) cannot be used to reveal consumers' attention to marketing communication activities. For example, statement 7 explores the importance of providing information but not paying attention to that information (they may consider it important to provide that information, but that doesn't mean the respondent actually pays attention to it).

Answer: We appreciate your valuable comments. Regarding the definition of marketing communication, we agree with your comment and, as such, we have made important changes throughout the text, particularly in section 2.1., which we expect to help clarify the concept of green marketing communication. Regarding the fact that the statements used in the questionnaire (especially statements 5 and 7) explore the importance of providing information, and not paying attention to the information, we understand the difference, but we assume that if the respondent considers certain information important, then he is necessarily attentive to it. When using the “think aloud” technique, this question did not raise doubts among the participants.

Comment: A big problem is the selection of a sample of respondents. Individual groups of respondents were unevenly represented (2/3 women, 52% of respondents under 31 years of age, more than half of the respondents were in the "until secondary" category). Probably the authors obtained the answers without random selection and also without respecting the population structure of Portugal. The conclusions cannot then be generalized at all. It is valid only in the specially created sample of respondents.

Answer: We appreciate your valuable comment. We agree with your criticism and, in accordance with it, we amended the text by adding the following text in the conclusion:

The present study contains some limitations. First, the sample is not entirely random and the survey took place online, thus excluding some members of the population who lack Internet access. Secondly, the survey was distributed on social networks, so the sample may also represent individuals who are more users of social networks, which may explain specific characteristics of the sample (e.g., age structure, educational structure). Therefore, we recommend some caution as regards any generalization of these results. “

Comment: For regression analysis “the coefficient of determination is R2 =0.45.” Is it a large enough coefficient to accept the proposed model? After all, only "45% of variation in green purchasing behavior can be explained through the consumers' attention".

Answer:

We appreciate your valuable comment. Although the coefficient is equal to 0.45, the model is globally relevant (observe the global significance of the model: F(1, 640) = 523.974, p-value = 0.000).

Comment: Also, some conclusions made in the discussion are problematic, e.g. “…. that labels and packaging would have another prominence and would be considered important for consumers as green marketing tools with the potential to influence consumer`s behavior. However, in this study, respondents attach little importance to this issue ….”

But according to Figure 1, half of respondents agreed with the statement "I pay close attention to product labels and tags with all the information about their environmental impacts" and almost 34% of respondents did not answer the question...

Answer: Thank you for your comment. We probably failed to convey clearly what we intended. In fact, according to Figure 1, half of respondents agreed with the statement "I pay close attention to product labels and tags with all the information about their environmental impacts". However, comparing with the remaining statements, it appears that the percentage of respondents who pay little attention to these aspects or consider them indifferent is relatively high. In order to clarify this point, the text has been changed as follows:

On the other hand, a considerable number of respondents (50.7%) assume not paying close attention to product labels and tags with all the information about their environmental impacts, or consider it indifferent/neutral.”

“This result stands out from the others since, in comparative terms with the remaining statements, it presents the highest percentage of respondents who strongly disagree, disagree or who assume a neutral position”

And the statement presented in the discussion was corrected:

 “However, in this study, respondents attach little importance or are indifferent to this issue (Statement 4, in the appendix).”

Reviewer 2 Report

My general comment on this article is that it shows 'only one side of the coin'.  Your research shows what people say  BEFORE they actually buy goods. (similar outcomes are known for other countries in Europe,  a 'political correct' majority all over the place). It does not show what they actually DO when buying stuff. Than they show much more 'selfish'behaviour. You only come across this when asking people when they leave shops (with the goods still in their hands. The interviewing has to be done by 'innocent students', not by 'officials. See for a full account of this in 'Adventures in EcoDesign of Electronic products' p271-279. By the way here you find (already, 1998) the statement that education, income and gender have a big influence on fundamental attitudes! The book is available for free through www.aeki.se.

The other aspect not addressed by you is that 'green'has quite a different meaning for different customer groups . This issue has been pioneered by Jacqueline Ottman in the USA (1995 and onwards) . Green can enhance sales in different ways for different groups : for price buyers (1/3 of total in Europe) it is the material reduction aspect; for 'tech buyers' also 1/3, it is the latest technology aspect (which is mostly 'greener) and for quality buyers (again 1/3) it 'positive emotion' like long life and recyclable (not energy saving because that is supposed to go at the cost of functionality. Tyhe communality of these three groups is that 'green has to be put into the persperctive of aspects which are superseding for the customer.! 

So saying: our products are so green, are so green will not work, neither will general education work (although many authors from academia believe so)

Author Response

To Reviewer 2:

First of all, we would like to thank you for all your contributions that allow us to considerably improve our research paper. Below, we present each of the changes made according to your comments:

Comment: Your research shows what people say BEFORE they actually buy goods (similar outcomes are known for other countries in Europe, a 'political correct' majority all over theplace). It does not show what they actually DO when buying stuff. Than they show much more 'selfish' behaviour. You only come across this when asking people when they leave shops(with the goods still in their hands. The interviewing has to be done by 'innocent students', not by 'officials. See for a full account of this in 'Adventures in EcoDesign of Electronic products' p271-279. By the way here you find (already, 1998) the statement that education, income and gender have a big influence on fundamental attitudes! The book is available for free through www.aeki.se.

Answer: Thank you for this observation. We believe that it is arguable that the way the survey was carried out could suggest that what respondents do. However, we based our investigation on previous studies that used similar scales for the behaviour variable (e.g., Moser, 2015; Lai and Cheng, 2016; Cardoso and Van Schoor, 2017; Uddin and Khan, 2018; Joshi and Rahman, 2019; Paço et al., 2019). We tried to analyze what respondents usually do (“I usually buy….”. This variable is different from the Intention variable, which the literature considers as readiness to buy (I am Wiiling to buy….) (e.g., Lai and Cheng, 2016);  an intention that may or may not translate into a buying behavior.                                                                                                      

       On the other hand, we recognize that some of the variables mentioned, such as education, income and gender, can influence attitudes. However, our study analyzes the influence of gender and education not on attitudes, but on attention to green marketing communication. In this sense, the present study will be able to contribute to a greater understanding of the relationships between these different variables.

Comment: The other aspect not addressed by you is that 'green' has quite a different meaning for different customer groups. This issue has been pioneered by Jacqueline Ottman in the USA (1995 and onwards). Green can enhance sales in different ways for different groups: for price buyers (1/3 of total in Europe) it is the material reduction aspect; for 'tech buyers' also 1/3, it is the latest technology aspect (which is mostly 'greener) and for quality buyers (again 1/3) it 'positive emotion' like long life and recyclable (not energy saving because that is supposed to go at the cost of functionality. The communality of these three groups is that 'green has to be put into the perspective of aspects which are superseding for the customer.!

So saying: our products are so green, are so green will not work, neither will general education work (although many authors from academia believe so)

Answer:

We appreciate your valuable comment. We understand that these interpretation differences may exist and that they will probably persist despite the efforts of environmental academic researchers. In order to clarify the concept of green and green products, in lines 197 to 204 we specify which green products we are referring to in this specific study. Moreover, in the questionnaires, the respondents were presented with the definition of green products as follows.

“We will now present you with a set of questions which main goal is to know your behavior in relation to environmentally friendly products, also known as green products. These products do not harm or have a reduced negative impact on the environment, that is, these products strive to protect or improve the natural environment by conserving energy and/or resources and reducing or eliminating the use of toxic agents, pollution, and waste”.

We based our definition in Ottman et al. (2006)

(Ottman, J.A., Stafford, E.R., Hartman, C.L., 2006. Avoiding green marketing myopia: ways to improve consumer appeal for environmentally preferable products. Environment, 48, 22-36.)

Reviewer 3 Report

Examining consumers' attitude to companies' green communication in the context of influencing their green purchasing behavior, as well as examining the importance of consumer characteristics in this attitude may be of great importance to the industry. The authors have raised several important questions that deserve extensive discussion and consideration. However, in order to emphasize the contribution of the research, and for the purpose of understanding its usefulness, authors are advised to refer to the following comments

1. General comments

Corrections are required regarding grammar, and the editing of sentences. It is important to use the right words for accurate presentation. If the text is improved, the message you intend to convey will be much clearer and more understandable. Some examples that require appropriate consideration are presented below.

a. The title. The composition of the sentences in the title creates unnecessary confusion. Authors are advised to reword the title concisely for greater coherence.

b.   Lines 56 - 61. The sentences are worded in a grammatically awkward way. It is recommended to emphasize the connection between them, through reformulation for better understanding.

c.    References. Some references are shown with capital letters and some are not. Authors are advised to present the references consistently according to accepted practice.

2. Specific comments

2.1. Abstract

   Using the concept of green communication may indicate energy-saving communication technologies, and this is obviously not the intention of this study. Authors are advised to define the term.

2.2 Introduction

a.    The use of the term "greener" can refer to sustainability in its various aspects while the authors referred to environmental sustainability in the study. Authors are advised to clarify which aspect they mean and to attach references for accuracy.

b. The introduction lacks the presentation of the importance of answering the research question. It is recommended that the authors expand on the need for research and how the results complement what is required.

2.3 Literature Review, Key Concepts and Research Hypotheses

a.  The authors use the concept of a socially responsible company without describing this concept. It is recommended to present the definition of the concept and the relevance to the research explicitly.

b.    The concept of an eco hotel requires an explanation since hotels may be considered as ecological service providers as well as buildings designed according to the standard of green building. It is recommended to explain the concept for better understanding.

c.    The presentation of the hypotheses requires explanations in relation to their importance and in relation to their exact definition. The surrounding background may be important. It is recommended that the authors explain the applicability of gender and whether the examination of consumers is based on the Portuguese context. If so, the answers may depend on local culture.

2.4 Discussion

a. This chapter includes a reference to the results that should be presented in a separate part.

b.  It is recommended that the authors present a segmentation of the results in terms of gender and education for the various products, which the authors referred to in the study, in view of the gender and cultural context. It is possible that certain products are more gender and culturally specific than others, for example building materials and cosmetics.

c.  The discussion lacks extensive reference to the local aspect of the research and its implications. It is recommended that the authors address this in the discussion in comparison to other cultures and then present the limitations of the study, according to their approach

Author Response

To Reviewer 3:

First of all, we would like to thank you for all your contributions that allow us to considerably improve our research paper. Below, we present each of the changes made according to your comments:

General Comments: Corrections are required regarding grammar, and the editing of sentences. It is important to use the right words for accurate presentation. If the text is improved, the message you intend to convey will be much clearer and more understandable. Some examples that require appropriate consideration are presented below. The title. The composition of the sentences in the title creates unnecessary confusion. Authors are advised to reword the title concisely for greater coherence. Lines 56 - 61. The sentences are worded in a grammatically awkward way. It is recommended to emphasize the connection between them, through reformulation for better understanding. References. Some references are shown with capital letters and some are not. Authors are advised to present the references consistently according to accepted practice.

Answer: We appreciate your valuable comment. Several corrections have been made to the text regarding grammar and sentence editing. We sincerely hope that our message will now be transmitted in a much clearer and more understandable way. Regarding the title, we hope that the new title will be more understandable: Analysing the influence of green marketing communication on consumers' green purchasing behaviour. With this new title, we hope to convey the objective of this research study, which consists of analysing companies' green communication and its influence on consumers' green purchase behaviour. Regarding the written English, we made several changes to improve the written English, including the text referring to lines Regarding lines 56-61. Finally, as for bibliographic references, they are now consistently presented in line with accepted practice.

Specific Comments:

Abstract:

Using the concept of green communication may indicate energy-saving communication technologies, and this is obviously not the intention of this study. Authors are advised to define the term.

Introduction:

The use of the term "greener" can refer to sustainability in its various aspects while the authors referred to environmental sustainability in the study. Authors are advised to clarify which aspect they mean and to attach references for accuracy. The introduction lacks the presentation of the importance of answering the research question. It is recommended that the authors expand on the need for research and how the results complement what is required.

Literature Review, Key Concepts and Research Hypotheses:

The authors use the concept of a socially responsible company without describing this concept. It is recommended to present the definition of the concept and the relevance to the research explicitly. The concept of an eco-hotel requires an explanation since hotels may be considered as ecological service providers as well as buildings designed according to the standard of green building. It is recommended to explain the concept for better understanding. The presentation of the hypotheses requires explanations in relation to their importance and in relation to their exact definition. The surrounding background may be important. It is recommended that the authors explain the applicability of gender and whether the examination of consumers is based on the Portuguese context. If so, the answers may depend on local culture

Discussion:

This chapter includes a reference to the results that should be presented in a separate part. It is recommended that the authors present a segmentation of the results in terms of gender and education for the various products, which the authors referred to in the study, in view of the gender and cultural context. It is possible that certain products are more gender and culturally specific than others, for example building materials and cosmetics. The discussion lacks extensive reference to the local aspect of the research and its implications. It is recommended that the authors address this in the discussion in comparison to other cultures and then present the limitations of the study, according to their approach.

Answer:

We appreciate your valuable comment regarding the abstract. We agree that the way the expression “green communication” was referred to in the abstract could lead to some misunderstandings, so we changed the expression in the abstract in order to avoid interpretation problems. Here is the corrected abstract:

“This study aims to explore whether consumers` attention to companies` green marketing communication influences their green purchase behaviour. It is also proposed to analyse the importance of consumers` characteristics, namely gender, education and green attitudes, in their attention to companies` green marketing communication. An online survey was carried out on the population residing in Portugal over 18 years of age, allowing to collect of 690 valid responses. Data analysis techniques were used, including descriptive analyses, parametric and non-parametric tests, linear correlation and regression analysis. The achieved results allow us to conclude that consumers are attentive to the companies' green marketing communication. It was also identified a strong correlation between consumers` attention to companies` green marketing communication and green purchasing behaviour. The results also confirm that individuals with higher educational levels, green attitudes, and females are the most attentive to companies` green marketing communication.”

Regarding the introduction, we have made considerable improvements to the test that, in our opinion, allow us to better understand the main objective of our study. We follow present the improved introduction:

Over the last decades, particularly since the industrial revolution, humanity’s rapidly growing consumption of natural resources is causing many environmental problems worldwide, affecting severely biodiversity and threatening human well-being (Li et al., 2019; Ramayah et al., 2010; Vlek and Steg, 2007; WWF, 2020; IPCC, 2021). As individual consumers, our behaviour is not environmentally benign, being partly responsible for the severity of all these environmental problems (Stern, 2000; DÄ…browska and JanoÅ›-KresÅ‚o, 2018). Nevertheless, the individuals` growing perception that the planet is already reaching very high levels of pollution has contributed to the rise of an environmental protection “movement” (Lee, 2009; Wiedmann et al., 2020). Furthermore both consumers and companies, particularly in more developed economies, are becoming increasingly aware of the urgent need to adopt more environmentally friendly consumer behaviour and production strategies (Cohen, 2014; White et al., 2019).

Purchasing green products for daily consumption is a good example of an environmentally responsible behaviour, capable of minimizing and solving many of the current environmental threats and, in recent years, it has been increasingly attracting the attention of both companies and consumers (Gonçalves et al., 2016; Biswas, 2017; Nguyen et al., 2019; Sheng et al., 2019; Urban et al., 2019). In addition to changing consumption patterns, by becoming greener, consumers also demand a more sustainable and environmentally responsible posture from companies. These havea key role as they develop and promote the products we consume and thus contribute to shaping demand and its associated environmental impacts. Over the last few decades, the role of sustainability in business has gradually increased, and several companies have significantly contributed to the promotion of sustainable consumption (UNIDO, 2018; Brozović et al., 2020; Pereira et al., 2021). Moreover, companies have increasingly adapted their activities within a more sustainable approach due to both stronger sustainability requirements and the implementation of new technologies aimed at improving the environmental, social, and economic business impacts (Brozović et al., 2020). Recently, Forbes (2019) selected and ranked large companies that excel in sustainability combined with high revenues, offering several key insights into the success of large companies in sustainability. For companies that want to assume a “greener” position, to show consumers how committed they are to contributing to a more sustainable planet, and also to gain a differentiating factor concerning competitors, environmentally responsible marketing, also known as green marketing, represents an unquestionable essential tool (Papadas et al., 2017). According to Paço et al. (2019), this tool is crucial, mainly when consumers show a low interest in certain products, including green ones.

If, on the one hand, research on green consumerism is evolving, justifying new research work that deepens knowledge in this area, green marketing is also an area for research due to its novelty and importance for environmental sustainability. The literature highlights the need to understand and promote consumer behaviour in green marketing and to identify the factors that influence green purchasing (Sharma, 2021). Tan et al. (2022) highlight that green marketing is essential to change the environmental behavior of consumers. According to the authors, companies should pay special attention to green communication, as it can increase customer confidence in a brand's environmental commitment and thus positively influence green purchasing. Therefore, in this research study, it is proposed to deepen, from the consumer's perspective, how attentive consumers are to companies' green marketing communication and how this influences consumers` green purchase decisions. Since the 1990s, the economic growth of Portugal, a country that joined the European Economic Community (EEC) in 1986, has been considerably increasing, which has ledding to several environmental problems worsening. As a European Union (EU) member, Portugal has been receiving monetary funds to accompany the other EU members in adopting environmental protection measures. A clear increase in Portuguese concerns for the main environmental issues is observed, but little is known on the consumers` perceptions regarding green companies and marketing communication (D`Souza et al., 2009; Nunes, 2019).

In this context, we propose to overcome this gap by studying Portuguese consumers` attention to companies` green marketing communication. To this end, an online survey was carried out on the population residing in Portugal over 18 years of age, allowing to collect of a total of 704 responses, of which 690 were considered valid and incorporated in the analysis.

              The remainder of this paper is organized as follows. After an introduction to the theme, a second section reviews the scientific literature, explores some of the main concepts and presents the research hypotheses. The following section is devoted to the methodology, with a detailed description of the questionnaire design, data collection and sample profile. Then, the results are presented and discussed. Finally, in the last section, the main conclusions are presented along with policy and research implications.

Regarding the Literature Review, Key Concepts and Research Hypotheses, we agree with your comments. In fact, throughout the text we refer to the concept of a socially responsible company without describing it. As such, we decided to adopt and write in the text the definition referred in the source: Vveinhardt J., Andriukaitiene R. Diagnostics of management culture in order to implement the concept of a socially responsible company: the case of a concern / J. Vveinhardt, R. Andriukaitiene // E&M — Ekonomie a Management. — 2016. — No. XIX(3). — Page 142157. As stressed by this source, socially responsible companies have a social obligation to carry out the policy in making decisions and acting in accordance with the accepted values of society. We have therefore added the following text:

 “(a company with the social obligation to carry out the policy in making decisions and acting in accordance with the accepted values of society (Vveinhardt and Andriukaitiene, 2016)).”

Regarding the concept of an eco-hotel, we fully agree with your comment. In fact, an eco-hotel can be considered environmentally friendly both for the service it provides and for the building that respects the environment. Hence, in lines 204 e 205, we defined, following ……., eco-hotels as: hotels that adopt green practices and/or whose buildings respect the environment.

Finally, regarding the discussion, we agree and appreciate your valuable comments. Our study was based on the study of behaviour considering globally the purchasing behaviour of green products. We recognize that the study of the purchase behaviour of specific green products could be an interesting field of future research, already indicated in the Conclusions and future research section. We also consider that future works may deepen the influence of contextual elements on the purchasing behaviour of green products, such as economic income or cultural issues. This aspect was also introduced in that section.

Reviewer 4 Report

It was a pleasure to review your text and I believe that the magazine will receive quality paper adequate for readers and suitable for future research. The structure of the paper is excellent and easy to read, the literature is adequately cited and the research was correctly conducted. Like most of the research being done online, unfortunately, this also does not represent a representative sample of Portugal. It would be correct to state that in the limitations of the work. You can maybe consider as limitation (or not?) educational structure, age structure and gender structure. It would be nice to see a research model.

Author Response

To Reviewer 4:

First of all, we would like to thank you for all your contributions that allow us to considerably improve our research paper.

Comment: It was a pleasure to review your text and I believe that the magazine will receive quality paper adequate for readers and suitable for future research. The structure of the paper is excellent and easy to read, the literature is adequately cited, and the research was correctly conducted. Like most of the research being done online, unfortunately, this also does not represent are presentative sample of Portugal. It would be correct to state that in the limitations of the work. You can maybe consider as limitation (or not?) educational structure, age structure and gender structure. It would be nice to see a research model

Answer: We appreciate your compliments. In fact, it was with great pleasure that we developed this scientific study on a topic that we like so much and that we believe to be of great importance in the current scenario of high concern for the environment. We agree with your advice and added some limitations in the conclusion section as presented below:

“The present study contains some limitations. First, the sample is not entirely random, and the survey took place online, thus excluding some members of the population who lack Internet access. Secondly, the survey was distributed on social networks, so the sample may also represent individuals who are more users of social networks, which may explain specific characteristics of the sample (e.g., age structure, educational structure). Therefore, we recommend some caution as regards any generalization of these results

Furthermore, Figure 1 with the research model was introduced.

Round 2

Reviewer 1 Report

The authors devoted a lot of effort towards improving their scientific publication. The improvement is evident. Therefore, I now recommend the article to be printed.

Reviewer 2 Report

Publish!